# The Role of Maternal and Early-Life Diet in Inflammatory Bowel Disease

**DOI:** 10.3390/nu16244292

**Published:** 2024-12-12

**Authors:** Katerina Karaivazoglou, Christos Triantos, Ioanna Aggeletopoulou

**Affiliations:** 1Department of Psychiatry, University Hospital of Patras, 26504 Patras, Greece; karaivaz@hotmail.com; 2Division of Gastroenterology, Department of Internal Medicine, University Hospital of Patras, 26504 Patras, Greece; chtriantos@upatras.gr

**Keywords:** IBD, colitis, maternal diet, early-life diet, intestinal inflammation

## Abstract

Background/Objectives: Inflammatory bowel disease (IBD) is a chronic gastrointestinal disorder with debilitating symptoms and multifactorial etiology. Nutritional factors during adult life have been implicated in IBD pathogenesis. In addition, there is growing evidence that maternal and early-life diet may be associated with intestinal inflammation and colitis severity. The aim of the current review was to detect and critically appraise all evidence regarding the role of maternal and early-life diet on intestinal inflammation. Methods: We performed a thorough search of the literature across two databases (Pubmed, ScienceDirect) using a variety of relevant terms. Results: A total of 23 studies, 16 experimental and 7 clinical, met inclusion criteria and were included in this review. Experimental studies reveal that high-fat and high-protein diets during gestation and neonatal life induce gut dysbiosis, amplify intestinal inflammation, and exacerbate colitis. In addition, a variety of nutritional factors included in maternal diets may affect offspring’s microbiota composition and intestinal health. Human studies concluded that maternal diet quality and the intake of fish and vegetables and of food fortified with vitamin D during gestation and early infancy significantly decreased IBD risk. However, human data are limited, and larger investigations are needed to further clarify the complex associations between specific nutritional compounds and intestinal inflammation. Conclusions: Dietary factors during pregnancy and early-life are involved in IBD pathogenesis, exerting either an exacerbating or protective effect. Improving pregnant women’s dietary habits could be a cost-effective strategy to reduce future IBD burden.

## 1. Introduction

Inflammatory bowel disease (IBD) is a group of chronic gastrointestinal disorders characterized by gut dysbiosis and intestinal mucosal inflammation. IBD patients commonly present with a variety of gastrointestinal symptoms including diarrhea, abdominal pain, rectal bleeding, and rectal urgency, which cause great impairment in everyday functioning and significantly disturb their quality of life [1]. Moreover, IBD is frequently characterized by weight loss and extra-intestinal manifestations affecting several organs such as the skin, the liver, the eyes, and the joints in the context of a more generalized systemic inflammation [1]. Crohn’s disease and ulcerative colitis represent the two main disease types of the IBD category, with both shared and differentiating characteristics [2]. Both disorders are characterized by chronic intestinal inflammation and the aforementioned constellation of clinical symptoms; however, in Crohn’s disease the inflammation may affect all the layers of the intestinal wall and any part of the gastrointestinal tract, while in ulcerative colitis, the inflammation is localized in the inner-most layer of the gut wall and predominantly affects the colon and the rectum [3]. In addition, distinct histopathological characteristics and differentiated patterns of genetic, immune, and environmental involvement have been identified regarding the two conditions [4].

According to large-scale epidemiological studies, several factors have been implicated in IBD pathophysiology including genetic predisposition, infections, lifestyle habits, and dietary factors [5]. At this point in the research, there are numerous studies focusing on the involvement of nutritional and dietary parameters in the etiology and clinical course of IBD, which mainly originate from adult clinical populations. These findings suggest that certain types of foods including red meat, sugar, and soft drinks are associated with increased risk of IBD [6,7], while the consumption of fruits and vegetables exerts a protective role against intestinal inflammation [8]. In addition, dietary interventions modulate intestinal immune response and may positively affect microbiota composition and function [9].

Intestinal homeostasis is crucial for human health and is largely based on the integrity of the gut barrier, which ensures the selective passage of useful nutrients and the blockade of toxic agents. Gut dysbiosis disrupts the integrity of the intestinal barrier, thus leading to bacterial translocation into the systemic circulation, which, in turn, triggers immune activation and leads to systemic inflammation [8]. In line with that, there is currently a growing understanding that deviances in microbiota gut colonization and early immune programming underlie the histopathological and pathophysiological alterations observed in IBD patients [10,11,12]. In this context, modern research has put particular emphasis on events occurring during the fetal, neonatal, and infant life and especially on the association between maternal and early-life diet and intestinal inflammation processes and has provided an increasing number of interesting data. Most relevant evidence is derived from animal models of intestinal inflammation and colitis; nonetheless, there is a limited number of studies involving human subjects. We decided to include both the maternal and early-life period in this review, given that they represent a developmental continuum that may have profound and long-lasting effects on health and homeostasis [13]. According to Acevedo et al., the pregnancy and the first year of life constitute the critical timeframe when environmental factors may interact with genetic predisposition through epigenetic mechanisms, thus representing either a window of opportunity (protective influences) or a window of susceptibility (risk influences) [14]. In this context, the aim of the current review is to detect and critically appraise all these findings in order to provide any clinically relevant novel information and formulate novel pathophysiological hypotheses to guide future research.

## 2. Methodology

We performed a meticulous search of the literature across Pubmed and ScienceDirect, using a variety of relevant terms in any possible combination in an attempt to extract as much information as possible (Table 1). The search strategy included a series of consecutive searches using each time two terms, one from the first and one from the second column of Table 1 connected with “and”, for example “inflammatory bowel disease AND maternal diet”, “inflammatory bowel disease AND early-life diet” and so on.

In addition, we conducted a manual search using all references of the selected articles to detect additional relevant research in the literature. The last search was performed on 31 July 2024. Inclusion criteria for the present review were (1) original articles published in English, (2) studies including inflammatory bowel disease patients or animal models of colitis, and (3) studies assessing the effect of dietary factors during pregnancy and the first two years of life (early-life) for humans and the first 4 weeks of life for animal experimental models (rodents). We decided to include not only studies on IBD patients but also research on colitis animal models, given that these preclinical studies have the potential to provide valuable information on the pathophysiology of intestinal inflammation, which is the hallmark of IBD. In addition, although breastfeeding represents an important element of early-life diet, studies reporting data on its role as a protective factor against IBD were not included in the current review, given that there is already a comprehensive meta-analysis [15] and a series of well-designed systematic reviews on this topic [16,17]. Our initial search yielded 420 titles and after excluding duplicates, 227 titles remained. Two researchers independently reviewed all abstracts, and 23 articles fulfilled inclusion criteria. Figure 1 contains a flow diagram that illustrates the articles’ selection process.

## 3. Studies’ Characteristics

A total of 7 studies [18,19,20,21,22,23,24] included human subjects while the remaining 16 were experimental studies [25,26,27,28,29,30,31,32,33,34,35,36,37,38,39,40] recruiting animal models of colitis (rodents). Among the human studies, four were case–control [18,20,23,24] and three were prospective cohort investigations [19,21,22]. One case–control study [24] included only Crohn’s disease patients, while the other three [18,20,23] included both Crohn’s disease and ulcerative colitis patients. In a similar way, one case–control study [24] included exclusively adult patients while the remaining three included either only pediatric patients [18,23] or youth patients under 25 years of age [20]. As far as the human prospective studies were concerned, in all of them participants were followed-up for a long time-period, ranging from 10 to 30 years. All the experimental studies were prospective and among them eight were randomized [26,28,32,33,35,36,37,38]. Table 2 provides an overview of all human and Table 3 all animal studies included in the current manuscript, examining dietary interventions and their effects on health outcomes, including study designs, dietary parameters, and participant characteristics.

## 4. Experimental Studies

High-fat diets have been strongly linked to increased bacterial translocation, microbiota changes, and intestinal inflammation. Based on this link, there has been a growing focus on the effects of high-fat diet during the gestational and infant period on the brain–gut axis and the structure and function of the enteric mucosa. Animal studies have consistently shown that maternal and early-life high-fat diet and being over-weight during the neonatal period are significantly associated with increased intestinal permeability, release of pro-inflammatory cytokines (IL-1β, IL-6, IL-17, TNF-α, IFN-γ), activation of intestinal inflammation, altered microbiota composition (intestinal dysbiosis), enhanced susceptibility to experimental colitis, and greater disease activity [25,26,27,28,31,40]. In addition, early modulation of cytokines’ secretion has led to normalization of intestinal permeability and reduced susceptibility to intestinal inflammation [19,28]. Maternal dietary fat composition and particularly the ratio of n-6/n-3 polyunsaturated fatty acids (PUFA) seems to play a crucial part in offspring’s responsiveness to experimental colitis [33,37]. N-3 PUFA exert their anti-inflammatory properties through cytokines’ suppression and reduction in AA-derived proinflammatory eicosanoids, while n-6 PUFA facilitates inflammatory processes via the synthesis of pro-inflammatory eicosanoids [37]. In line with that, an earlier randomized study showed that increased dietary intake of n-6 PUFA by pregnant mice led to increased susceptibility to colitis and more severe intestinal inflammation in their offspring, while increased intake of n-3 PUFA was associated with preventive and therapeutic effects [33]. However, another study by the same research group concluded that lower levels of n-6 PUFA in maternal diet resulted in reduced colonic crypt depth and increased intestinal permeability in their offspring. According to these authors, n-6 PUFA plays an important role in the maturation of the colon and the integrity of the intestinal barrier through the production of prostaglandin E2 (PGE2) [32]. These contradictory research findings regarding the role of PUFA in the pathophysiology of intestinal inflammation, indicate that complex interactions are implicated, and further research is needed. In an attempt to clarify these discrepancies, a more recent randomized investigation examined the effect of several maternal diets with different concentrations of n-6 and n-3 fatty acids [37] on offspring’s susceptibility to colitis. That study showed that the n-6 PUFA diet was associated with more severe inflammation and exacerbated colitis symptoms, while the n-3 PUFA diet and the combined n-6/n-3 PUFA diet were linked with milder colitis and a reduction in colitis-associated inflammation indices [37]. Moreover, it was revealed that a balanced n-6/n-3 PUFA diet during gestation and neonatal life may be more effective in lowering pro-inflammatory cytokines’ levels and protecting against experimentally induced colonic damage compared to a n-3 PUFA diet [37]. The authors attributed their findings to the fact that there are shared enzymes implicated in the metabolism of n-6 and n-3 PUFA and for this reason, the dominance of either type of fatty acids would disturb the other’s conversion, leading to an increase in inflammatory molecules [37]. Furthermore, apart from the effect of dietary fat on the emergence of colitis, there is research assessing the effect of high-protein intake on gut flora and intestinal inflammation [35,41]. According to a recent randomized study, maternal high-protein diet during gestation and lactation significantly decreased microbiota diversity and amplified disease activity in mice offspring compared to maternal high-fat and control diets [35]. In addition, the observed effects were more pronounced in offspring that continued to be fed the same diet after weaning [35]. Nutritional factors, including PUFA, have been linked to immune programming and inflammation pathophysiology through epigenetic mechanisms, namely DNA methylation, histone modifications, and miRNA expression [14]. Most relevant research has focused on allergies, diabetes, and obesity [42]. Given that IBD is characterized by gut dysbiosis and immune dysregulation, it would be useful to determine whether the effects of early-life nutrition on intestinal pathophysiology are mediated by epigenetic pathways.

Food contains bioactive compounds, which exert anti-inflammatory effects, regulate the immune system, and protect against the development of several chronic diseases including IBD [43]. In a recent investigation, early-life diet containing high concentrations of broccoli sprouts was associated with less severe disease activity and increased microbiota richness in an immune model of Crohn’s disease [30]. Broccoli sprouts are rich in sulforaphane, which suppresses inflammatory signaling and the observed benefits, especially in terms of microbiota composition and histological damage, were mostly evident in younger mice [30]. In a similar way, pediatric diets high in cellulose protected against chemically induced colitis [36]. These effects were mainly attributed to cellulose’s trophic action, which leads to increased colonic length and elongation of enteric crypts, thus attenuating the toxic effects of colitis-inducing agents [36]. However, the observed benefits were transient, and it is not clear whether they would be reproduced in immune-mediated models of colitis [36]. In contrast to the protective action of the aforementioned compounds, there are also findings revealing that certain dietary interventions including prebiotic supplementation may worsen intestinal inflammation. Galacto-oligosaccharides and inulin during gestation appeared to increase inflammation indices and exacerbate clinical symptomatology and histopathological damage in mice offspring, mainly through alterations in gut microbiota and in colonic lipid content and gene expression [29,34]. In addition, according to Schaible et al. [38], the supplementation of maternal diet with methyl donors including betaine, choline, vitamin B12, and folate acid, amplified offspring’s susceptibility to colitis, possibly through increased expression of immune-related genes, decreased expression of the *cpn2* gene, which protects against inflammation, and specific colitis-inducing alterations in gut microbiota. In contrast, an identical pediatric post-weaning diet did not affect young adults’ mice’s responsiveness to colitis-inducing chemical agents. These findings suggest that gestation is the critical period during which the colon appears most vulnerable to toxic insults. In addition, although methyl donor deficiency has been associated with experimental colitis [44], research has shown that excessive methyl donor intake may have detrimental effects on health outcomes [45]. Given that certain methyl donors are routinely used as maternal diet supplementation compounds in humans, these results should be further explored through more extended animal and human studies in order to optimize maternal diet supplementation policy. Finally, an earlier experimental study [39] on the effect of dietary phytoestrogens during gestation failed to reveal any protective effect against colonic inflammation but instead showed that phytoestrogens may induce a more severe inflammatory response in colitis models.

## 5. Clinical Studies

Epidemiological studies have shown great variations in the prevalence of IBD world-wide with industrialized Western populations presenting greater percentage of either Crohn’s disease or ulcerative colitis compared with populations originating from less developed regions [46]. In addition, there is research revealing that migration from developing to Western countries is associated with increased IBD risk [47,48], while the rapid industrialization of developing countries during the current century has led to a rise in IBD prevalence rates [49]. Among the factors that might explain these variations are dietary habits that range considerably between industrialized and non-industrialized societies.

In this context, there is emerging research focusing on the role of nutritional factors during gestation and infant life on the incidence of IBD. A recent case–control investigation across three geographical regions showed that Crohn’s disease patients consumed greater amounts of ultra-processed and processed food from infancy to adolescence compared not only to healthy unrelated controls but also to healthy first-degree relatives [24]. These results indicate that the role of early-life diet in IBD pathogenesis might act synergistically to the effect of genetic factors, given that Crohn’s disease patients’ first-degree relatives are expected to be genetically predisposed to the emergence of CD. Additionally, a previous study across nine countries reported that the regular intake of vitamins, iron, and other minerals during pregnancy was associated with a lower risk of Crohn’s disease and ulcerative colitis [20]. Earlier case–control studies that assessed a variety of potential risk factors reported conflicting results regarding the effect of gestational and neonatal factors [18,20,23]. For example, an Italian case–control study in pediatric IBD patients concluded that the introduction of gluten prior to 6 months of age was significantly associated with an increased risk of ulcerative colitis and Crohn’s disease [23]. However, earlier investigations did not reveal any association between the age of introduction of various foods including flour and pediatric IBD risk [18,20]. Likewise, while Baron et al. [18] reported that the consumption of sucrose-supplemented milk during childhood was associated with the diagnosis of ulcerative colitis, another previous study had failed to detect any association between sucrose-supplemented milk during infancy and IBD risk [20].

Given that the aforementioned findings were based on retrospective collection of data, prospective studies were designed attempting to clarify the contribution of early-life dietary factors on IBD risk. A recent study prospectively assessed the effect of alcohol intake and certain traditional food consumption during pregnancy in the Faroese population where IBD is particularly prevalent [50]. That study failed to reveal any significant associations of maternal dietary parameters with IBD risk [50]. Although it had a 30-year follow-up, it included a relatively small cohort originating from a restricted geographical region, which might not allow for statistically significant correlations to emerge [50]. In contrast, in a recent Scandinavian study that prospectively followed up two large cohorts of children until adolescence and young adulthood, the overall quality of diet at 1 year of age was significantly associated with IBD risk [21]. In addition, high fish and vegetable intake at that same age protected against the later development of IBD, while the consumption of sugar-sweetened beverages predicted increased IBD risk [21]. At 3 years of age, only fish intake was associated with a later IBD diagnosis and especially UC, suggesting that the effect of dietary factors on intestinal health and immune programming takes place quite early in life. Finally, another Scandinavian cohort study with a 30-year follow-up revealed that in utero exposure to small extra doses of vitamin D modestly reduced IBD risk [19]. The authors suggested that greater prenatal doses of dietary vitamin D might lead to greater reduction of IBD risk and emphasized the health value of vitamin D supplementation policies, especially in higher latitude regions with lower exposure to sunlight.

## 6. Conclusions and Future Directions

In conclusion, there are solid experimental data that gestational and neonatal dietary factors may have major effects on intestinal inflammatory processes. More specifically, high-fat and high-protein maternal and early-life diet alters gut microbiota, induces the release of inflammatory agents, and aggravates colitis symptomatology. Likewise, the ratio of dietary n-6 and n-3 PUFA during pregnancy is implicated in early immune programming and gut health, and a balanced PUFA diet may protect against gut dysbiosis and intestinal inflammation. In addition, regarding the role of specific dietary bioactive compounds in IBD pathogenesis, sulphoraphane and cellulose appear to protect against colonic inflammation while certain prebiotics, methyl donors, and phytoestrogens may be associated with increased colitis susceptibility. However, these data on the effects of specific bioactive compounds on intestinal health are less conclusive and should be further corroborated with larger experimental and clinical trials in order to inform therapeutic and preventive interventions. Additionally, in addition to the aforementioned experimental data, there is also a limited number of relevant human studies that need to be replicated and expanded. According to their findings, the quality of diet during gestation and early infancy was strongly associated with IBD prevalence and could represent a modifiable risk factor. Moreover, most experimental and clinical studies concluded that dietary factors exerted their effects on intestinal inflammation mainly during the gestational period, while these effects tended to fade-out or even vanish during the later stages of infant life. In this respect, pregnancy constitutes a key time-period during which proper dietary interventions might prove most beneficial in terms of reducing IBD risk. Improving pregnant women’s dietary habits and increasing the consumption of high-quality unprocessed food would constitute a feasible, efficacious, and cost-effective public health strategy to prevent IBD, thus alleviating the current disease burden. In Figure 2, the impact of maternal and early-life dietary factors on gut health and IBD development is depicted.

In total, human data are still sparse and need to be replicated through wider-scale case–control and prospective cohort studies. Experimental models of colitis are extremely useful in studying IBD pathophysiology but may not fully capture the complex nature of the immunological mechanisms underlying IBD. Furthermore, the effect of maternal and early-life diet should be explored in populations with a variety of lifestyle and dietary habits, given that existing findings mainly derive from the Scandinavian countries. For example, it would be extremely interesting to explore the impact of the Mediterranean, Asian, African, and plant-based diets during pregnancy on IBD risk. Providing solid, indisputable evidence might shift consumers’ eating choices and turn a part of the food industry towards more health-promoting products.

## Figures and Tables

**Figure 1 nutrients-16-04292-f001:**
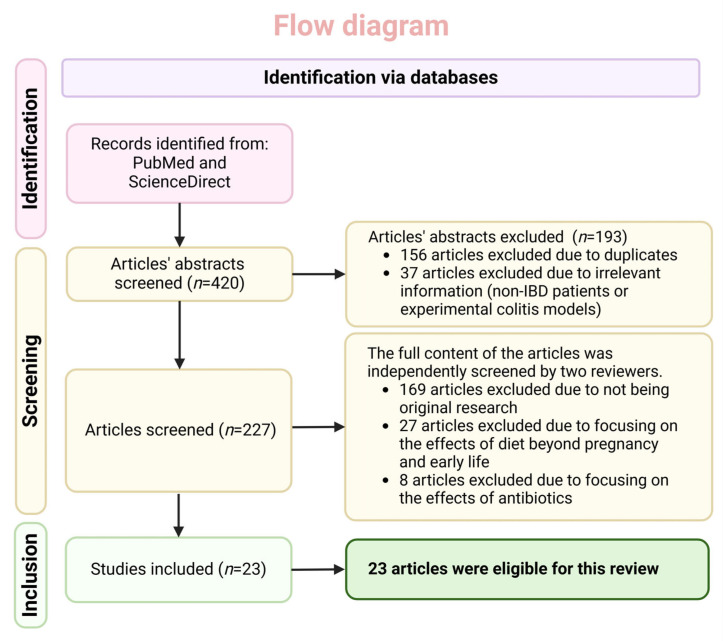
Flow diagram illustrating the article selection process. Created with BioRender.com (accessed on 13 November 2024).

**Figure 2 nutrients-16-04292-f002:**
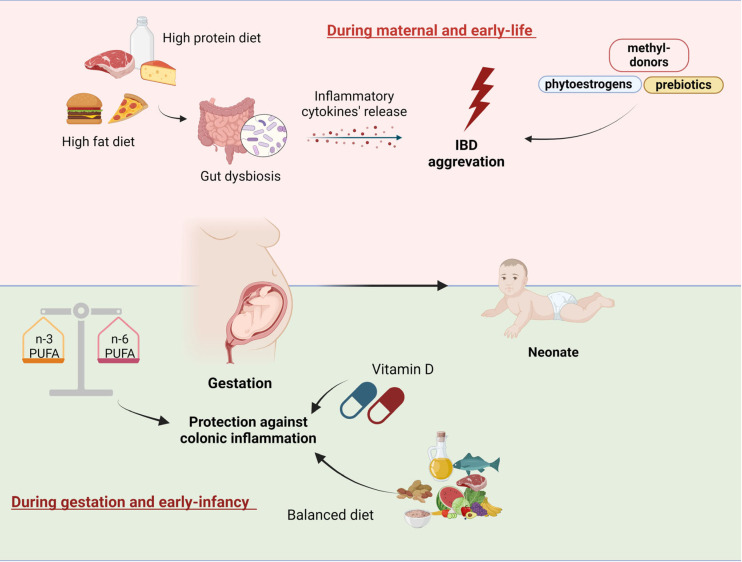
Maternal and early-life dietary influences on the risk of inflammatory bowel disease development and protection mechanisms. The **top** panel illustrates how a maternal high-fat and high-protein diet can induce gut dysbiosis, promoting inflammatory cytokine release, which may aggravate IBD in early life. Bioactive compounds such as phytoestrogens, methyl donors, and prebiotics may modulate this risk. The **bottom** panel highlights how balanced maternal intake of n-3 and n-6 polyunsaturated fatty acids (PUFAs) and adequate vitamin D levels during gestation and early infancy can provide protection against colonic inflammation in the neonate, suggesting the importance of a balanced maternal diet during pregnancy and lactation for reducing IBD risk in offspring. Created with BioRender.com (accessed on 14 October 2024).

**Table 1 nutrients-16-04292-t001:** Search terms used in this review.

Inflammatory Bowel Disease Synonyms and Relevant Terms	Maternal and Early-Life Diet Synonyms and Relevant Terms
Inflammatory bowel disease, IBD, Crohn’s disease, ulcerative colitis, intestinal inflammation	Maternal diet, early-life diet, gestational dietary factors, gestational nutritional factors

**Table 2 nutrients-16-04292-t002:** Overview of human studies assessing dietary interventions and their impact on health outcomes.

Author	Ref	Publication Year	Country	Type of Study and Participants	Design	Dietary Parameters
Guo et al.	[21]	2024	Sweden/Norway	81,280 participants followed-up until adolescence and young adulthood. Food intake data were collected at 1 and 3 years of age	Prospective cohort	Food intake assessed with a questionnaire and diet quality assessed with the Healthy Eating Index
Trakman et al.	[24]	2022	Australia/China/USA	274 CD pts, 82 first-degree relatives, 83 household members, and 92 non-cohabiting healthy unrelated controls. Retrospective assessment of early-life processed food intake	Case–control cross-sectional observational	Validated dietary questionnaire to assess early-life processed food intake
Duus et al.	[19]	2021	Denmark	Two birth cohorts: 103,606 individuals born from 1 June 1983 to 31 May 1985 exposed to vit D fortification vs. 113,643 individuals born from 1 September 1986 to 31 August 1988 not exposed, followed-up for 30 years	Prospective cohort	Diet fortified with vitamin D vs. control diet
Hammer et al.	[22]	2019	Denmark/Faroe	6 cohorts with 5698 (mothers and their newborn babies) followed-up for 10 to 31 years	Observational prospective cohort	Self-report questionnaire assessing dietary intake of pilot whale/blubber and fish
Strisciuglio et al.	[23]	2017	Italy	264 pediatric IBD pts vs. 203 controls. Retrospective assessment of infant diet	Case–control	Infant diet assessed with the use of a parent-reported questionnaire
Baron et al.	[18]	2005	France	222 pediatric CD pts and 60 pediatric UC pts vs. 1:1 matched controls. Retrospective assessment of infant diet	Population-based, matched case–control	Infant diet assessed with parent interviews by blinded investigators
Gilat et al.	[20]	1987	Israel	302 young CD pts and 197 young UC pts diagnosed before 20 years of age vs. 1:2 matched controls either healthy or with another chronic diseases. Retrospective assessment of neonatal and infant diet	Case–control	Neonatal and early infancy diet assessed with a parent interview

Abbreviations: CD, Crohn’s disease; pts, patients; vs., versus; vit D, vitamin D; IBD, inflammatory bowel disease; UC, ulcerative colitis.

**Table 3 nutrients-16-04292-t003:** Overview of experimental animal studies assessing dietary interventions and their impact on health outcomes.

Author	Ref	Publication Year	Country	Type of Study and Participants	Design	Dietary Parameters
Holcomb et al.	[30]	2023	USA	IL-10-KO mice, 4 mice fed the control diet vs. 5 mice fed the treatment diet starting at 4 wks of age and lasted for 3 wks. Inoculation with Helicobacter hepatitis to trigger CD-like symptoms at 5 wks of age	Prospective controlled	Diet containing 10% raw broccoli sprouts vs. control diet
Huang et al.	[31]	2023	China	C57BL/6 female mice fed HF/HS vs. control diet during gestation and lactation and their offspring fed HF/HS vs. control diet post-weaning for 4 wks. CD-like colitis was induced at 8 wks of age (TNBS)	Randomized prospective controlled	High-fat/high-sugar vs. control diet
Le et al.	[34]	2023	France	BALB/cJRj pregnant mice fed standard diet vs. diet with GOSs and inulin for 5 weeks; 8–10 week old offspring received colitis-inducing DSS	Prospective controlled	Maternal diet with prebiotic galacto-saccharides and inulin vs. control diet
He et al.	[29]	2022	China	16 Sprague Dawley female rats fed a fiber-free vs. inulin diet. Their 8 week-old offspring received DSS to induce colitis	Prospective controlled	Inulin diet vs. fiber-free diet
Liu et al.	[35]	2022	China	C57BL/6 female pregnant mice fed HF vs. HP vs. control diet during pregnancy and lactation. Their 8 week-old offspring received either colitis-inducing DSS or water	Randomized prospective controlled	Maternal high-fat vs. maternal high-protein vs. maternal control diet
Al Nabhani et al.	[25]	2019	France	C57BL/6 male mice pups exposed to normal vs. early-life HF diet vs. adult life HF diet. At 20 weeks of age mice received either colitis-inducing DSS or water	Prospective controlled	High-fat vs. normal diet
Xie et al.	[40]	2018	China	C57BL/6 female mice fed HF vs. control diet during pregnancy and lactation. Their post-weaning offspring were fed normal diet and at 8 wks of age they received colitis-inducing DSS	Prospective controlled	High-fat vs. control diet
Bibi et al.	[26]	2017	USA	C57BL/6J female mice fed HF vs. control diet during pregnancy and lactation. After weaning, 9 female offspring from either group were fed HF diet and at 14 weeks of age they received colitis-inducing DSS	Randomized prospective controlled	High-fat vs. control diet
Gulhane et al.	[28]	2016	Australia	3 week-old Winnie mice fed HF vs. control diet for 9 weeks	Blind randomized prospective controlled	High-fat vs. normal control diet
Reddy et al.	[37]	2016	India	24 Wistar female rats randomly assigned to 4 equal groups: n-6 vs. low n-3 vs. n-6/n-3 vs. n-3 diet; 20 offspring per group were fed the same diet and received colitis-inducing DSS or water at 35 days of age	Randomized prospective controlled	n-6 vs. low n-3 vs. n-6/n-3 vs. n-3 PUFA diet
Gruber et al.	[27]	2015	Germany/UK/The Netherlands	Female WT mice fed HF vs. control diet during pregnancy and only HF diet during lactation. WT and ARE (genetically predisposed to colitis) offspring were fed HF or control diet and were assessed at 8–12 weeks of age	Prospective controlled	High-fat vs. control diet
Nagy-Szakal et al.	[36]	2013	USA	C57BL/6J young mice, 10 fed HC vs. 10 fed LC diet for 60–90 days and then received colitis-inducing DSS	Randomized prospective controlled	High- vs. low-cellulose diet
Schaible et al.	[38]	2011	USA	C57BL/6 female mice randomly assigned to MD (10 mice) vs. control (10 mice) diet during pregnancy and lactation. Their offspring received colitis-inducing DSS)	Randomized prospective controlled	Methyl donor vs. control diet
Innis et al.	[32]	2010	Canada	Sprague Dawley female rats randomly assigned to sunflower oil (SO) vs. canola oil (CO) vs. 20% fish oil (20% FO) vs. 10% fish oil diet. At 3 months their offspring received colitis-inducing DNBS	Randomized prospective controlled	Sunflower oil (SO) vs. canola oil (CO) vs. 20% fish oil (20% FO) vs. 10% fish oil (10% FO) diet
Seibel et al.	[39]	2008	Germany	Female Wistar rats fed PE-enriched vs. PE-depleted diet during pregnancy and lactation. Male post-weaning offspring fed PE-depleted (n = 16) vs. PE-enriched (n = 10) diet, received colitis-inducing DSS at 11 weeks of age	Prospective controlled	Phytoestrogen-enriched vs. phytoestrogen-depleted diet
Jacobson et al.	[33]	2005	Canada	Female Sprague Dawley rats randomly assigned to high 18:3n-3 vs. high 18:1n-9 vs. high 18:2n-6 fatty acids diet during pregnancy and lactation. Their offspring received colitis-inducing DNBS at 15 days of age	Randomized prospective controlled	High 18:3n-3 vs. high 18:1n-9 vs. high 18:2n-6 fatty acids diet

Abbreviations: IL-10, interleukin 10; KO, knock out; vs., versus; IBD, inflammatory bowel disease; pts, patients; PUFA, polyunsaturated fatty acids; HF, high-fat; HP, high-protein; HC, high-cellulose; LC, low-cellulose; PE, phytoestrogen; DNBS, 2,4-dinitrobenzene sulfonic acid; DSS, extraneous sulphate sodium; MD, methyl donor; GOSs, galacto-saccharides.

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
