# Peer review of "The Role of Maternal and Early-Life Diet in Inflammatory Bowel Disease"

_nutrients, 2024, doi:10.3390/nu16244292_

Round 1

Reviewer 1 Report

Comments and Suggestions for Authors

The manuscript entitled “The role of maternal and early-life diet in inflammatory bowel disease” presents interesting issue but there are serious problems.

Major:

(1)    The manuscript is prepared as a systematic review, as Authors suggest systematic literature search, but they do not specify if they prepared a systematic review or not. Authors should be aware that systematic review presents a key role in broadening knowledge, but the other reviews do not present such role, so preparing a subjective review may not be valuable contribution.

(2)    It seems that Authors combined 2 totally different issues – maternal diet and early life diet, which is not clearly justified. Both factors may influence IBD development, but it is not sufficient to combine them in common study (as there are also other factors). It should be clearly described why did Authors decide to present those issues together.

(3)    Authors prepare a review (we do not know if it is systematic or not), while other authors already conducted a systematic review for the included issues, namely Agrawal, M.; Sabino, J.; Frias-Gomes, C.; Hillenbrand, C.M.; Soudant, C.; Axelrad, J.E.; Shah, S.C.; Ribeiro-Mourão, F.; Lambin, T.; Peter, I.; et al. Early life exposures and the risk of inflammatory bowel disease: Systematic review and meta-analyses. EClinicalMedicine 2021, 36, 100884, doi:10.1016/j.eclinm.2021.100884. Moreover, the mentioned systematic review included also meta-analysis, which causes that it is even more valuable. Interestingly, Agrawal et al. in 2021 (3 years ago) included to their review more studies than Authors of the presented manuscript do in 2024. So the question arises why? Is it associated with a highly subjective inclusion to the presented study? We must be aware that it is impossible to have less studies published in 2024 than it was in 2021. We rather suppose that the study wrote in 2024 should be broader. Meanwhile a vast majority of the studies included to a presented manuscript were also presented in the systematic review from 2021, while novel studies were not included.

(4)    It may be noted, that novel and important studies are not included and not presented within the review, as the cohort study by Agrawal et al, published in 2024, presenting the effect of breastfeeding on the IBD development (Agrawal M, Størdal K, Vinkel Hansen A, Östensson M, Brusco De Freitas M, Allin KH, Jess T, Ludvigsson J, Mårild K. Breastfeeding Duration Is Not Associated With Offspring Inflammatory Bowel Disease Risk in Three Population-Based Birth Cohorts. Clin Gastroenterol Hepatol. 2024 May 9:S1542-3565(24)00416-6. doi: 10.1016/j.cgh.2024.04.013. Epub ahead of print. PMID: 38729392.)

(5)    Last but not least, the manuscript is extremely shabbily prepared and even the references are not clearly referred, eg:

·         For the study presented in the main body of the study as Huang et al [13], the reference [13] may be found as: Gilat, T.; Hacohen, D.; Lilos, P.; Langman, M.J. Childhood factors in ulcerative colitis and Crohn's disease. An international cooperative study. Scand J Gastroenterol 1987, 22, 1009-1024, doi:10.3109/00365528708991950.

·         For the study presented in the main body of the study as Le et al [14], the reference [14] may be found as: Guo, A.; Ludvigsson, J. Early-life diet and risk of inflammatory bowel disease: a pooled study in two Scandinavian birth cohorts. Gut 2024, 73, 590-600, doi:10.1136/gutjnl-2023-330971.

The indicated studies are only 2 random examples while the number of such problems is much higher.

For sure the indicated references are not those that should be indicated. For a review it is crucial to properly describe and refer the studies as without it the study is of no value. We do not know what studies are here described and it is even hard to assess the value of the presented review.

Author Response

Reviewer 1

Comment #1

“The manuscript is prepared as a systematic review, as Authors suggest systematic literature search, but they do not specify if they prepared a systematic review or not. Authors should be aware that systematic review presents a key role in broadening knowledge, but the other reviews do not present such role, so preparing a subjective review may not be valuable contribution

Response: Thank you for your comment. This is not a systematic review; however, we tried to perform a methodical and thorough search of the literature. We are fully aware that systematic reviews provide high-quality evidence-based information, however we strongly believe that every well-written review may contribute to the existing knowledge. This is an invited manuscript, and the editors had agreed for a plain review on this particular topic.

Comment #2

“ It seems that Authors combined 2 totally different issues – maternal diet and early life diet, which is not clearly justified. Both factors may influence IBD development, but it is not sufficient to combine them in common study (as there are also other factors). It should be clearly described why did Authors decide to present those issues together.”

Response: Thank you for your comment. There are many studies addressing the effects of maternal and early-life diet on health outcomes together, since the gestational and neonatal period constitute a critical developmental continuum with significant long-term impacts for a variety of chronic conditions. For example, in their comprehensive systematic review and meta-analysis, Agrawal et al (2021) included both maternal and neonatal exposures. In this respect, we consider maternal and early-life diet as two distinct yet closely interconnected parameters. We have added relevant clarification on this issue in our manuscript (p. 2, lines:68-74).

Comment #3

Authors prepare a review (we do not know if it is systematic or not), while other authors already conducted a systematic review for the included issues, namely Agrawal, M.; Sabino, J.; Frias-Gomes, C.; Hillenbrand, C.M.; Soudant, C.; Axelrad, J.E.; Shah, S.C.; Ribeiro-Mourão, F.; Lambin, T.; Peter, I.; et al. Early life exposures and the risk of inflammatory bowel disease: Systematic review and meta-analyses. EclinicalMedicine 2021, 36, 100884, doi:10.1016/j.eclinm.2021.100884. Moreover, the mentioned systematic review included also meta-analysis, which causes that it is even more valuable. Interestingly, Agrawal et al. in 2021 (3 years ago) included to their review more studies than Authors of the presented manuscript do in 2024. So the question arises why? Is it associated with a highly subjective inclusion to the presented study? We must be aware that it is impossible to have less studies published in 2024 than it was in 2021. We rather suppose that the study wrote in 2024 should be broader. Meanwhile a vast majority of the studies included to a presented manuscript were also presented in the systematic review from 2021, while novel studies were not included.”

Response: Thank you for your comment. The systematic review and meta-analysis by Agrawal focused on a wide range of early-life exposures (for example maternal smoking and health, environmental pollutants, perinatal factors, neonatal infections, breastfeeding, hygiene-related factors, antibiotics etc) and not exclusively on diet and this is the reason it includes more studies. Furthermore, in that review the only dietary parameter that was included was breastfeeding and this is why we did not include breastfeeding studies, and we focused on other nutritional factors to avoid extensive overlap and provide useful information.

Comment #4

It may be noted, that novel and important studies are not included and not presented within the review, as the cohort study by Agrawal et al, published in 2024, presenting the effect of breastfeeding on the IBD development (Agrawal M, Størdal K, Vinkel Hansen A, Östensson M, Brusco De Freitas M, Allin KH, Jess T, Ludvigsson J, Mårild K. Breastfeeding Duration Is Not Associated With Offspring Inflammatory Bowel Disease Risk in Three Population-Based Birth Cohorts. Clin Gastroenterol Hepatol. 2024 May 9:S1542-3565(24)00416-6. Doi: 10.1016/j.cgh.2024.04.013. Epub ahead of print. PMID: 38729392.)”

Response: Thank you for your comment. As already mentioned in the preceding response and in our manuscript (p.3, lines 95-98), we decided not to include studies on the effects of breastfeeding, given that there are already comprehensive and high-quality systematic reviews and meta-analysis on this topic.

Comment #5

but not least, the manuscript is extremely shabbily prepared and even the references are not clearly referred, eg:

  • For the study presented in the main body of the study as Huang et al [13], the reference [13] may be found as: Gilat, T.; Hacohen, D.; Lilos, P.; Langman, M.J. Childhood factors in ulcerative colitis and Crohn's disease. An international cooperative study. Scand J Gastroenterol 1987, 22, 1009-1024, doi:10.3109/00365528708991950.
  • For the study presented in the main body of the study as Le et al [14], the reference [14] may be found as: Guo, A.; Ludvigsson, J. Early-life diet and risk of inflammatory bowel disease: a pooled study in two Scandinavian birth cohorts. Gut 2024, 73, 590-600, doi:10.1136/gutjnl-2023-330971.

The indicated studies are only 2 random examples while the number of such problems is much higher.

For sure the indicated references are not those that should be indicated. For a review it is crucial to properly describe and refer the studies as without it the study is of no value. We do not know what studies are here described and it is even hard to assess the value of the presented review.”

Response: Thank you for your comment. The mistakes in the reference list are due to different formatting. We apologize for that, and we have corrected it in the manuscript.

Reviewer 2 Report

Comments and Suggestions for Authors

With interest, I read the manuscript nutrients-3287074.

In principle, the die behind this work is good, however, substantial improvements are required for it to get publishable.

  1. The article needs to be somewhat restructured. First of all, from the introduction, the details of the literature search should be moved to sections presenting methodology (new section) and general results (could be added to “2. Studies’ characteristics“).
  2. Second, a real introduction much be substantially expanded. For example, the Authors never explain what ulcerative colitis and Crohn disease are … (differentiation, specific features). What is IBD in general is also not explained. Symptoms, pathophysiology, etc.
  3. The methodology must be much more detailed. I understand that only original studies were included, while reviews, commentaries, editorials not? Why from 227 articles only 23 were selected – Figure 1 does not give any answer. Please make it much more detailed and describe thoroughly in the methodological or results sections, where appropriate. Overlap PubMed/ScienceDirect, etc.?
  4. Table 1. Please, provide exact phrases used for the search with connectors, brackets, etc., so that the Reader could reproduce your search if required.
  5. What was the date of search?
  6. Table 2 is not acceptable. First of all, it should be divided into two tables, one for human and one for animal studies. Then, their content should be much expanded: so far only some (no all required, see further) methodological details are given, but not output. The tables should provide the output of the reported studies. Characteristics/features of the animal models, groups numbers of animals or human participants should be given, readouts as well.
  7. Several substances are mentioned in the text, e.g. PUFA, the mechanisms of supplementation with which have been examined in general or in context of other diseases, e.g. allergies. The effects of several of those have been found to be epigenetically mediated (PMID: 33668787). Those mechanisms must be reported and discussed in the context of the studies on IBD reported here. Furthermore, are the concepts like “window of opportunity“ (and other “windows“; PMID: 33668787) know to Authors? Are they applicable in the context of IBD?
  8. Please, be more detailed. E.g. why “methyl donors“. What is their importance?
  9. Why HFD effect is so widely studied in IBD in the perinatal context?

Author Response

Reviewer 2

Comment #1

“The article needs to be somewhat restructured. First of all, from the introduction, the details of the literature search should be moved to sections presenting methodology (new section) and general results (could be added to “2. Studies’ characteristics“)”

Response: Thank you for your constructive comment. It has been done (p.2, line 78).

Comment #2

Second, a real introduction much be substantially expanded. For example, the Authors never explain what ulcerative colitis and Crohn disease are … (differentiation, specific features). What is IBD in general is also not explained. Symptoms, pathophysiology, etc.”

Response : The reviewer is right. Further information has been added (p.1, lines 32-43; p.2, lines 44-46)

Comment #3

The methodology must be much more detailed. I understand that only original studies were included, while reviews, commentaries, editorials not? Why from 227 articles only 23 were selected – Figure 1 does not give any answer. Please make it much more detailed and describe thoroughly in the methodological or results sections, where appropriate. Overlap PubMed/ScienceDirect, etc.?”

Response:

The reviewer is right. We have enriched the flow diagram to provide more information regarding the study selection process.

Comment #4

“Table 1. Please, provide exact phrases used for the search with connectors, brackets, etc., so that the Reader could reproduce your search if required.”

Response : Thank you for your comment. It has been added (p.2, lines 81-84).

Comment #5

“What was the date of search?”

Response : Thank you for your comment. It has been added (p.3, line 87).

Comment #6

“Table 2 is not acceptable. First of all, it should be divided into two tables, one for human and one for animal studies. Then, their content should be much expanded: so far only some (no all required, see further) methodological details are given, but not output. The tables should provide the output of the reported studies. Characteristics/features of the animal models, groups numbers of animals or human participants should be given, readouts as well.”

Response: The reviewer is right. We divided table 2 into two tables according to the reviewer’s suggestions. In addition, we enriched the information provided in those tables. However, we decided not to include results since we comprehensively reported them in the text.

Comment #7

“Several substances are mentioned in the text, e.g. PUFA, the mechanisms of supplementation with which have been examined in general or in context of other diseases, e.g. allergies. The effects of several of those have been found to be epigenetically mediated (PMID: 33668787). Those mechanisms must be reported and discussed in the context of the studies on IBD reported here. Furthermore, are the concepts like “window of opportunity“ (and other “windows“; PMID: 33668787) know to Authors? Are they applicable in the context of IBD?”

Response : Thank you for your  insightful comment. Relevant information has been added to the manuscript (p.2, lines 70-74; p.9, lines 189-194).

Comment #8

«Please, be more detailed. E.g. why “methyl donors“. What is their importance?»

Response: Thank you for your constructive remark. It has been done (p.10, lines 214-217 and 220-222).

Comment #9

«Why HFD effect is so widely studied in IBD in the perinatal context?»

Response : Thank you for your comment. It has been addressed (p.9, lines 145-148).

Reviewer 3 Report

Comments and Suggestions for Authors

Line 39 - Line 44

Please explain the link between gut bacteria, the intestinal immune response, and gut homeostasis.

Line 96 - Line 99

The references listed here do not match the number in the 'reference part', so please double-check and correct them.

Line 106 - Line 108

Which cytokines play an important role in maintaining intestinal homeostasis, and in what ways?

Line 156 - Line 161 

Is it because maternal supplementation with these agents induced gut histopathological damage of the offspring, or does it cause alterations in the gut flora of both the mother and the offspring? Kindly describe briefly.

Line 170 - Line 219

This section is too lengthy; please break it down into pieces. Include the association between IBD and dietary habits (such as maternal diets, early-life diets, and nutritional supplements).

Author Response

Reviewer 3

Comment #1

Line 39 – Line 44

Please explain the link between gut bacteria, the intestinal immune response, and gut homeostasis.”

Response: Thank you for your constructive comment. It has been done (p.2, lines 56-60)

Comment #2

“Line 96 – Line 99

The references listed here do not match the number in the ‘reference part’, so please double-check and correct them.”

Response : The reviewer is right. References have been properly formatted

Comment #3

Line 106 – Line 108

Which cytokines play an important role in maintaining intestinal homeostasis, and in what ways?

Response : Thank you for your comment. It has been added (p.9, line 151).

Comment #4

Line 156 – Line 161

Is it because maternal supplementation with these agents induced gut histopathological damage of the offspring, or does it cause alterations in the gut flora of both the mother and the offspring? Kindly describe briefly.”

Response : Thank you for your remark. Proper clarification has been added (p.10, lines 214-217).

Comment #5

“Line 170 – Line 219

This section is too lengthy; please break it down into pieces. Include the association between IBD and dietary habits (such as maternal diets, early-life diets, and nutritional supplements).”

Response: Thank you for your constructive comment. We divided the section into 3 paragraphs according to your suggestions.

Round 2

Reviewer 1 Report

Comments and Suggestions for Authors

The manuscript entitled “The role of maternal and early-life diet in inflammatory bowel disease” presents interesting issue but there are serious problems.

Major:

(1)    The manuscript does not present a systematic literature search, so its value is minor. Authors should be aware that systematic review presents a key role in broadening knowledge, but the other reviews do not present such role, so preparing a subjective review may not be valuable contribution.

(2)    It seems that Authors are not familiar with IBD, as they present intermittent nausea and vomiting among the most important symptoms, but they totally ignore major symptoms of the disease.

(3)    It seems that Authors combined 2 totally different issues – maternal diet and early life diet, which is not clearly justified. Both factors may influence IBD development, but it is not sufficient to combine them in common study (as there are also other factors). It should be clearly described and justified within the manuscript why did Authors decide to present those issues together.

(4)    Authors indicate that they “decided not to include studies on the effects of breastfeeding”, which causes that their study presents even more surprising scope – they decided to combine 2 totally different issues – maternal diet and early life diet, but not to present breastfeeding being within early life diet.

(5)    Authors prepare a review (not a systematic review but highly subjective review), while other authors already conducted a systematic review for the included issues, namely Agrawal, M.; Sabino, J.; Frias-Gomes, C.; Hillenbrand, C.M.; Soudant, C.; Axelrad, J.E.; Shah, S.C.; Ribeiro-Mourão, F.; Lambin, T.; Peter, I.; et al. Early life exposures and the risk of inflammatory bowel disease: Systematic review and meta-analyses. EClinicalMedicine 2021, 36, 100884, doi:10.1016/j.eclinm.2021.100884. Moreover, the mentioned systematic review included also meta-analysis, which causes that it is even more valuable. Interestingly, Agrawal et al. in 2021 (3 years ago) included to their review more studies than Authors of the presented manuscript do in 2024. So the question arises why? Is it associated with a highly subjective inclusion to the presented study? We must be aware that it is impossible to have less studies published in 2024 than it was in 2021. We rather suppose that the study wrote in 2024 should be broader. Meanwhile a vast majority of the studies included to a presented manuscript were also presented in the systematic review from 2021, while novel studies were not included.

Author Response

Reviewer 1

Comment #1

The manuscript does not present a systematic literature search, so its value is minor. Authors should be aware that systematic review presents a key role in broadening knowledge, but the other reviews do not present such role, so preparing a subjective review may not be valuable contribution.

Response: Thank you for your comment. This is an invited manuscript, and the editor has agreed on a plain review

Comment #2

It seems that Authors are not familiar with IBD, as they present intermittent nausea and vomiting among the most important symptoms, but they totally ignore major symptoms of the disease”

Response: Thank you for your comment. It has been corrected (p.1, lines 34 and 36).

Comment #3

“It seems that Authors combined 2 totally different issues – maternal diet and early life diet, which is not clearly justified. Both factors may influence IBD development, but it is not sufficient to combine them in common study (as there are also other factors). It should be clearly described and justified within the manuscript why did Authors decide to present those issues together.”

Response: The systematic review and meta-analysis by Agrawal et al (2021) on early-life exposures that the reviewer has mentioned in other comments, also included in utero and early-life exposures in the same paper. We believe that we provide adequate justification on this issue in our manuscript (p.2, lines 68-74).

Comment #4

Authors indicate that they “decided not to include studies on the effects of breastfeeding”, which causes that their study presents even more surprising scope – they decided to combine 2 totally different issues – maternal diet and early life diet, but not to present breastfeeding being within early life diet.”

Response: Thank you for your comment. We decided not to include breastfeeding because there is already a recent meta-analysis and 2 systematic reviews on this topic (p.3, lines 95-98).

Comment #5

Authors prepare a review (not a systematic review but highly subjective review), while other authors already conducted a systematic review for the included issues, namely Agrawal, M.; Sabino, J.; Frias-Gomes, C.; Hillenbrand, C.M.; Soudant, C.; Axelrad, J.E.; Shah, S.C.; Ribeiro-Mourão, F.; Lambin, T.; Peter, I.; et al. Early life exposures and the risk of inflammatory bowel disease: Systematic review and meta-analyses. EClinicalMedicine 2021, 36, 100884, doi:10.1016/j.eclinm.2021.100884. Moreover, the mentioned systematic review included also meta-analysis, which causes that it is even more valuable. Interestingly, Agrawal et al. in 2021 (3 years ago) included to their review more studies than Authors of the presented manuscript do in 2024. So the question arises why? Is it associated with a highly subjective inclusion to the presented study? We must be aware that it is impossible to have less studies published in 2024 than it was in 2021. We rather suppose that the study wrote in 2024 should be broader. Meanwhile a vast majority of the studies included to a presented manuscript were also presented in the systematic review from 2021, while novel studies were not included.”

Response: The systematic review by Agrawal et al (2021) is not limited to nutritional factors but includes other exposures such as social factors, perinatal factors, passive smoking, hygiene-related factors and others and this is why it includes more studies. Moreover, our review includes several studies that are not included in Agrawal’s review for example Guo (2024), Holcomb (2023), Huang (2023), Le (2023), He (2022), Liu (2022), Trakman (2022), Duus (2021).

Reviewer 2 Report

Comments and Suggestions for Authors

Thank you for addressing my comments. I have no further reservations.

Author Response

Comments and Suggestions for Authors

Thank you for addressing my comments. I have no further reservations.

We thank Reviewer 2 for his valuable comments.